# The spin Drude weight of the spin-1/2 $XXZ$ chain: An analytic finite size study

Andreas Klümper[1] and Kazumitsu Sakai[2*]

**1** Fakultät für Mathematik und Naturwissenschaften,
Bergische Universität Wuppertal, 42097 Wuppertal, Germany
**2** Department of Physics, Tokyo University of Science,
Kagurazaka 1-3, Shinjuku-ku, Tokyo 162-8601, Japan

\* k.sakai@rs.tus.ac.jp

April 25, 2019

## Abstract

**The Drude weight for the spin transport of the spin-1/2 $XXZ$ Heisenberg chain in the critical regime is evaluated exactly for finite temperatures. We combine the thermodynamic Bethe ansatz with the functional relations of type $Y$-system satisfied by the row-to-row transfer matrices. This makes it possible to evaluate the asymptotic behavior of the finite temperature spin Drude weight with respect to the system size. As a result, the Drude weight converges to the results obtained by Zotos (Phys. Rev. Lett. 82, 1764 (1999)), however with very slow convergence upon increase of the system size. This strong size dependence may explain that extrapolations from various numerical approaches yield conflicting results.**

# 1  Introduction

The Heisenberg spin-1/2 $XXZ$ chain is one of the most fundamental and extensively investigated models describing low-dimensional quantum magnetism. The model is exactly solvable due to the existence of infinitely many nontrivial local conserved charges. The static properties, such as the energy spectrum, thermodynamic quantities and even correlation functions, can be exactly calculated by various versions of the Bethe ansatz (BA). In contrast, the evaluation of quantities related to dynamical properties is in general much more difficult.

Nevertheless, by utilizing the integrability, several transport coefficients of the model have been calculated within linear response theory. For instance, for the thermal transport, the energy current itself is conserved [1], and hence the thermal Drude weight, which is the weight of a delta function contribution to the thermal conductivity, can be evaluated [2–4] by a generalized Gibbs ensemble.

The spin transport is a more intricate topic as the spin current – in contrast to the thermal current – is not a conserved quantity. The spin Drude weight at finite temperatures is the main topic of this paper. For certain anisotropy values of the $XXZ$ chain the finite temperature spin Drude weight was evaluated using exact means by Zotos [5] 20 years ago. These calculations consist of two steps. First, the spin Drude weight is expressed in terms of the thermal expectation value of the curvature of the energy eigenvalues with respect to the magnetic flux through the system. This is a finite-temperature generalization of the Kohn formula [6]. Second, by taking into account as carefully as possible the behavior of string type solutions to the Bethe ansatz equations with respect to the flux, the spin Drude weight is described by a set of non-linear integral equations that may be viewed as extended thermodynamic Bethe ansatz (TBA) equations [5]. Originally this procedure has been developed for the evaluation of the Drude weight for the Hubbard chain [7]. It is based on the so-called string hypothesis according to which bound states are described by equidistantly distributed Bethe rapidities with corrections that are mostly but not always exponentially small for large system sizes.

The results of [5] indicate that the Drude weight of the $XXZ$ chain in the critical regime is finite and monotonously decreases with increase of temperature as well as with increase of the anisotropy parameter. It is exactly zero at any finite temperature for the isotropic point and for the massive regime.

There exist, however, several controversies about the spin Drude weight for the $XXZ$ chain: various analytical or numerical studies yield contradictory results [8–17]. In recent years, novel approaches [18–24] have been developed based on the Mazur inequality, new conserved quasi-local charges, and their charge-charge and current-charge correlation functions. These can be calculated exactly by the Bethe ansatz or, at high temperatures, by more elementary means. Specifically at high temperatures, an optimal lower bound has been evaluated [19] which agrees with the high-temperature asymptotics of Zotos' results [5, 11]. For general temperature the extended TBA equations of [5] can be derived on the basis of

an optimal lower bound [24] complementing the derivation by the generalized hydrodynamics developed recently in [25–29]. Interestingly, the spin Drude weight at any finite temperatures exhibits a fractal dependence on the anisotropy parameter of the model [18, 19, 24].

Motivated by these results, we revisit this long-standing problem for the spin Drude weight of the $XXZ$ chain. We develop a completely different method for evaluating Kohn's formula. We avoid the treatment of string corrections by use of the $T$- and $Y$-systems [30] which are the functional relations satisfied by transfer matrices ($T$-functions) and certain combinations thereof ($Y$-functions). All excitations are described by real excitation parameters, i.e. the zeros of the $T$-functions. The curvature of the energy levels can be expressed by the $Y$-functions, and their particular zeros characterizing the state. By insertion of the distribution of zeros, which describes the thermal equilibrium, the thermal expectation value of the curvature of the energy spectrum and hence the Drude weight is obtained. In the thermodynamic limit, indeed the formulation of Zotos is obtained. However, our approach allows to estimate the large-size asymptotics of the spin Drude weight. We find that the Drude weight strongly depends on the system size, and very slowly converges to the result in the thermodynamic limit.

The layout of this paper is as follows. In the next section, we briefly formulate the spin transport within linear response theory. Also a finite-temperature generalization of the Kohn formula is presented. In section 3, we give the $T$- and $Y$-system defined for the six-vertex model which is the classical counterpart of the $XXZ$ chain. In section 4, we show that the energy for any excited state is expressed as the solution to the non-linear integral equations (NLIEs) satisfied by the $Y$-functions. The finite-temperature extension is described in section 5. The analysis of the NLIEs and the temperature and size-dependences of the Drude weight are presented in section 5. The last section is devoted to a summary and discussions. Some technical details are deferred to several appendices.

## 2 Drude weight

Let us consider the spin transport property of the spin-1/2 $XXZ$ chain on the periodic lattice with sites labeled by $k = 1, 2, \ldots, L$:

$$H = J \sum_{k=1}^{L} \left( \sigma_{k+1}^{+} \sigma_{k}^{-} + \sigma_{k}^{+} \sigma_{k+1}^{-} + \frac{\Delta}{2} \sigma_{k}^{z} \sigma_{k+1}^{z} \right). \tag{2.1}$$

Here $\sigma_k^x$, $\sigma_k^y$, $\sigma_k^z$ are the Pauli matrices acting on the $k$th site and $\sigma_k^{\pm} := (\sigma_k^x \pm \mathrm{i}\sigma_k^y)/2$. For later convenience, we parameterize the anisotropy parameter $\Delta$ as

$$\Delta = \cos\gamma \quad (0 \leq \gamma \leq \frac{\pi}{2}) \tag{2.2}$$

and set

$$\nu = \frac{\pi}{\gamma} \quad (\nu \geq 2). \tag{2.3}$$

To consider the spin conductivity $\sigma_{\mathrm{s}}$ within linear response theory, we utilize the Kubo formula [31, 32]

$$\sigma_{\mathrm{s}} := \lim_{\omega \to 0} \mathrm{Re}\,\sigma_{\mathrm{s}}(\omega), \quad \sigma_{\mathrm{s}}(\omega) := \lim_{\varepsilon \to +0} \frac{1}{L} \int_0^\infty \mathrm{d}t\, e^{-\mathrm{i}(\omega - \mathrm{i}\varepsilon)t} \int_0^\beta \mathrm{d}\lambda\, \langle \mathcal{J}_{\mathrm{s}}(-t - \mathrm{i}\lambda)\mathcal{J}_{\mathrm{s}} \rangle, \tag{2.4}$$

where $\langle\cdots\rangle$ denotes the thermal expectation value, $\beta$ is the reciprocal of temperature, $\beta = 1/T$, and $\mathcal{J}_\mathrm{s}$ is the total spin current given by

$$\mathcal{J}_\mathrm{s} = \sum_{k=1}^{L} j_k^\mathrm{s} = -\mathrm{i}J \sum_{k=1}^{L} (\sigma_{k+1}^+ \sigma_k^- - \sigma_k^+ \sigma_{k+1}^-), \tag{2.5}$$

which is naturally derived from the continuity equation:

$$\dot{S}_k^z = -(j_{k+1}^\mathrm{s} - j_k^\mathrm{s}), \quad \dot{S}_k^z = \frac{1}{2}\dot{\sigma}_k^z = \mathrm{i}\left[H, \frac{1}{2}\sigma_k^z\right]. \tag{2.6}$$

After some simple manipulation, the spin conductivity can be expressed as

$$\mathrm{Re}\,\sigma_\mathrm{s}(\omega) = \pi D\delta(\omega) + \sigma_\mathrm{s}^\mathrm{reg}(\omega). \tag{2.7}$$

The singular part $\pi D\delta(\omega)$ denotes the ballistic (dissipationless) contribution, whereas $\sigma_\mathrm{s}^\mathrm{reg}(\omega)$ describes the normal (dissipative) transport. The weight $D$ of the delta function is called the spin Drude weight and is a characteristic of ballistic transport. In the thermodynamic limit $L \to \infty$, (2.4) is reduced to

$$D := -\frac{1}{L}\left[\langle K \rangle + 2 \sum_{\substack{m,n \\ E_m \neq E_n}} p_n \frac{|\langle m|\mathcal{J}_\mathrm{s}|n\rangle|^2}{E_m - E_n}\right], \tag{2.8}$$

where $p_n := \mathrm{e}^{-\beta E_n}/(\sum_m \mathrm{e}^{-\beta E_m})$ is the Boltzmann weight for the energy eigenvalue $E_n$ corresponding to the eigenstate $|n\rangle$, and $\langle K \rangle$ is the thermal expectation value of the kinetic energy term $K := J\sum_k (\sigma_{k+1}^+ \sigma_k^- + \sigma_k^+ \sigma_{k+1}^-)$.

In general, the direct evaluation of (2.8) is a formidable task especially for $L \gg 1$. Fortunately, this expression can be transformed into a more accessible form. Let us introduce a flux $\phi$ into the system (2.1), i.e.

$$H(\phi) = J\sum_{k=1}^{L} \left(\mathrm{e}^{\mathrm{i}\phi/L}\sigma_{k+1}^+ \sigma_k^- + \mathrm{e}^{-\mathrm{i}\phi/L}\sigma_k^+ \sigma_{k+1}^- + \frac{\Delta}{2}\sigma_k^z \sigma_{k+1}^z\right), \tag{2.9}$$

and perturbatively expand the energy $E_n(\phi)$ for (2.9) in terms of $\phi/L$:

$$E_n(\phi) = E_n - \frac{\phi}{L}\langle n|\mathcal{J}_\mathrm{s}|n\rangle - \left(\frac{\phi}{L}\right)^2 \sum_{\substack{m,n \\ E_m \neq E_n}} \frac{|\langle m|\mathcal{J}_\mathrm{s}|n\rangle|^2}{E_m - E_n} - \frac{1}{2}\left(\frac{\phi}{L}\right)^2 \langle n|K|n\rangle + o\left(\phi^2/L^2\right). \tag{2.10}$$

Thus one finds that the Drude weight $D$ is identical to the thermal average of energy level curvatures with respect to $\phi$:

$$D = \lim_{L\to\infty} L \sum_n p_n \frac{\mathrm{d}^2 E_n(\phi)}{\mathrm{d}\phi^2}\bigg|_{\phi=0}. \tag{2.11}$$

This formula is the finite-temperature generalization of Kohn's formula [6].

## 3 $T$- and $Y$-systems

As shown in (2.11), to evaluate the finite-temperature Drude weight, we must carefully take into account the thermal expectation value of the second derivative of the energy with respect to the flux. To achieve this systematically, we translate the original problem for the (1+1)-dimensional quantum system into a problem for the two-dimensional classical lattice system. The lattice system corresponding to the $XXZ$ chain is the six-vertex model whose weights are given by the following six non-zero elements of the $R$-matrix $R(v) \in \mathrm{End}(V \otimes V)$ ($V$ denotes the two-dimensional vector space $\mathbb{C}^2$ spanned by the spin-up state $|0\rangle = \binom{1}{0}$ and the spin-down state $|1\rangle = \binom{0}{1}$):

$$R_{11}^{11}(v) = R_{22}^{22}(v) = \frac{[v+2]}{[2]}, \quad R_{12}^{12}(v) = \mathrm{e}^{\mathrm{i}\frac{\phi}{L}}\frac{[v]}{[2]}, \quad R_{21}^{21}(v) = \mathrm{e}^{-\mathrm{i}\frac{\phi}{L}}\frac{[v]}{[2]}, \quad R_{12}^{21}(v) = R_{21}^{12}(v) = 1, \tag{3.1}$$

where $[v] := \sin(\gamma v/2)/\sin\gamma$. The indices can be interpreted as

$$R(v)|\alpha\rangle \otimes |\beta\rangle = \sum_{\gamma,\delta} R_{\alpha\beta}^{\gamma\delta}(v)|\gamma\rangle \otimes |\delta\rangle. \tag{3.2}$$

The $R$-matrix satisfies the Yang-Baxter equation (YBE)

$$R_{23}(v)R_{13}(u)R_{12}(u-v) = R_{12}(u-v)R_{13}(u)R_{23}(v), \tag{3.3}$$

where $R_{jk}(v)$ acts on $V_j \otimes V_k$. Note that $V_j$ means the copy of $\mathbb{C}^2$ spanned by the $j$th state $|0\rangle_j$ and $|1\rangle_j$. The YBE guarantees that the family of row-to-row transfer matrices $T_1(v) \in \mathrm{End}(V^{\otimes L})$ constructed by

$$T_1(v) = \mathrm{Tr}_{V_0}\left[R_{0L}(i(v+\mathrm{i}))R_{0L-1}(\mathrm{i}(v+\mathrm{i}))\cdots R_{01}(\mathrm{i}(v+\mathrm{i}))\right] \tag{3.4}$$

consists of mutually commuting matrices for arbitrary spectral parameters $u$ and $v$:

$$[T_1(u), T_1(v)] = 0. \tag{3.5}$$

The original quantum system (2.9) can be expressed as the logarithmic derivative of $T_1(v)$ with respect to $v$:

$$H(\phi) = \frac{A}{2\pi\mathrm{i}}\frac{\partial}{\partial v}\log T_1(v)\Big|_{v=-\mathrm{i}} - \frac{JL}{2}\Delta, \quad A := \frac{4\pi J \sin\gamma}{\gamma}. \tag{3.6}$$

The transfer matrices $T_1(v)$ can be diagonalized by standard Bethe ansatz techniques. The resultant eigenvalues (simply denoted by $T_1(v)$) explicitly read

$$T_1(v) = \varphi(v-\mathrm{i})\frac{q(v+2\mathrm{i})}{q(v)}\mathrm{e}^{\mathrm{i}\frac{\phi}{2}} + \varphi(v+\mathrm{i})\frac{q(v-2\mathrm{i})}{q(v)}\mathrm{e}^{-\mathrm{i}\frac{\phi}{2}},$$

$$\varphi(v) := \left(\frac{\mathrm{sh}\,\frac{\gamma}{2}v}{\sin\gamma}\right)^L, \qquad q(v) := \prod_{k=1}^{m}\mathrm{sh}\,\frac{\gamma}{2}(v-v_k). \tag{3.7}$$

Here we have dropped an overall phase factor $\mathrm{i}^L \mathrm{e}^{\mathrm{i}\phi(m/L-1/2)}$ in the expression of $T_1(v)$. The $m \in \{0, 1, 2, \ldots, L/2\}$ unknown numbers $\{v_k\}$ in (3.7) are to be determined from the Bethe ansatz equation (BAE):

$$\frac{\varphi(v_k+\mathrm{i})}{\varphi(v_k-\mathrm{i})} = -\mathrm{e}^{\mathrm{i}\phi}\frac{q(v_k+2\mathrm{i})}{q(v_k-2\mathrm{i})}. \tag{3.8}$$

Instead of solving the BAE (3.8) directly, we introduce for convenience a more general family of transfer matrices ($T$-functions) including the above $T_1(v)$. Let us consider the $T$-functions of the form [30]

$$T_{n-1}(v) := \sum_{j=1}^{n} e^{\frac{i\phi}{2}(n-2j+1)} \varphi(v + i(2j - n - 1)) \frac{q(v + in)q(v - in)}{q(v + i(2j - n - 2))q(v + i(2j - n))}. \quad (3.9)$$

The following relations ($T$-system) are directly proven for any $v \in \mathbb{C}$ and integers $1 \leq j \leq n$,

$$T_{n-1}(v + ij)T_{n-1}(v - ij) = T_{n+j-1}(v)T_{n-j-1}(v) + T_{j-1}(v + in)T_{j-1}(v - in), \quad (3.10)$$

where we set $T_{-1}(v) = 0$ and $T_0(v) = \varphi(v)$.

From now on, for simplicity, we restrict ourselves to $\nu \in \mathbb{Z}_{\geq 2}$ in (2.3). In this case, the following relation holds

$$T_\nu(v) = 2(-1)^m \cos\left(\frac{\nu\phi}{2}\right) T_0(v + i\nu) + T_{\nu-2}(v). \quad (3.11)$$

Combining the $T$-functions, we construct the $Y$-functions [30]:

$$Y_j(v) = \frac{T_{j-1}(v)T_{j+1}(v)}{T_0(v + i(j+1))T_0(v - i(j+1))} \quad \text{for } 1 \leq j \leq \nu - 2,$$

$$Y_{\nu-1}(v) = (-1)^m e^{-i\frac{\nu\phi}{2}} \frac{T_{\nu-2}(v)}{T_0(v + i\nu)},$$

$$Y_\nu(v) = (-1)^m e^{i\frac{\nu\phi}{2}} \frac{T_{\nu-2}(v)}{T_0(v + i\nu)}. \quad (3.12)$$

Here we set $Y_0(v) = 0$. Using the $T$-system (3.10) and (3.11), one can easily find that the above relations are equivalent to the following

$$1 + Y_j(v) = \frac{T_j(v + i)T_j(v - i)}{T_0(v + i(j+1))T_0(v - i(j+1))} \quad \text{for } 1 \leq j \leq \nu - 2,$$

$$(1 + Y_{\nu-1}(v))(1 + Y_\nu(v)) = \frac{T_{\nu-1}(v + i)T_{\nu-1}(v - i)}{T_0(v + i\nu)T_0(v - i\nu)}. \quad (3.13)$$

Moreover we notice that the $Y$-functions (3.12) satisfy the functional relations ($Y$-system),

$$Y_j(v + i)Y_j(v - i) = (1 + Y_{j-1}(v))(1 + Y_{j+1}(v)) \quad \text{for } 1 \leq j \leq \nu - 3,$$
$$Y_{\nu-2}(v + i)Y_{\nu-2}(v - i) = (1 + Y_{\nu-3}(v))(1 + Y_{\nu-1}(v))(1 + Y_\nu(v)) \quad \text{for } \nu \geq 3,$$
$$Y_{\nu-1}(v + i)Y_{\nu-1}(v - i) = e^{-i\nu\phi}(1 + Y_{\nu-2}(v)),$$
$$Y_\nu(v + i)Y_\nu(v - i) = e^{i\nu\phi}(1 + Y_{\nu-2}(v)). \quad (3.14)$$

In the next section, we transform the above functional relations (3.14) into the non-linear integral equations (NLIEs) which determine the energy eigenvalues of the $XXZ$ model (2.9).

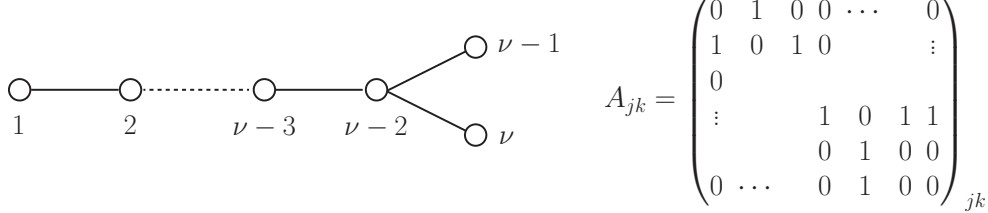

Figure 1: The adjacency matrix $A_{jk}$ for type $D_\nu$.

## 4  NLIEs for arbitrary excitations

From now on we consider the case that the system size is $L \in 4\mathbb{Z}$ and the number of BAE roots is $m = L/2 = 2\mathbb{Z}$.

The transfer matrix $T_1(v)$ (3.4) includes the Hamiltonian of the $XXZ$ model (2.9) via (3.6). The information on the energy spectrum is embedded in analytical properties of the $T$- and $Y$-functions. For instance, the ground state is characterized by those $T$-functions for which none of the $T_j(v)$ ($j = 1, \ldots, \nu - 1$) has any zero in the strip $\text{Im}\, v \in [-1, 1]$ (which we call the physical strip). On the other hand, any excited state is characterized by additional zeros $\zeta_j^k$ satisfying the conditions $\text{Im}\, \zeta_j^k \in [-1, 1]$ and

$$Y_j(\zeta_j^k + \mathrm{i}) = -1 \quad (1 \le k \le m_j \in \mathbb{Z}_{\ge 0};\ 1 \le j \le \nu), \tag{4.1}$$

or equivalently, from (3.13)

$$T_j(\zeta_j^k) = 0 \quad (1 \le j \le \nu - 1), \quad T_{\nu-1}(\zeta_\nu^k) = 0. \tag{4.2}$$

We give several remarks about the zeros $\{\zeta_j^k\}$. First, according to usual conventions we may call the zeros $\{\zeta_j^k\}$ ($1 \le j \le \nu-1$) "holes" as they correspond to solutions of BA like equations, but differ from BA rapidities. Second, by some numerical analysis and the form of $T_{\nu-1}(v)$ in (3.9), we find that $\{\zeta_\nu^k\}$ coincide with the real parts of the odd-strings which are BA rapidities whose imaginary parts are $\nu\mathrm{i}$. From the second equation in (3.13) we see that both $\{\zeta_{\nu-1}^k\}$ ($1 \le k \le m_{\nu-1}$) and $\{\zeta_\nu^k\}$ ($1 \le k \le m_\nu = m_{\nu-1}$) are zeros of $T_{\nu-1}(v)$. For $\phi = 0$, these zeros are degenerate: $\{\zeta_{\nu-1}^k\} = \{\zeta_\nu^k\}$. Finally, from (4.1) we obtain

$$-\log Y_j(\zeta_j^k + \mathrm{i}) = 2\pi\mathrm{i} I_j^k \quad (I_j^k \in \mathbb{Z} + 1/2). \tag{4.3}$$

For this reason, the functions $-\log Y_j(v + \mathrm{i})/(2\pi\mathrm{i})$ are interpreted as counting functions.

To find the solutions of (3.14), we adopt the following procedure (see appendix A for a detailed derivation). First we take the logarithmic derivative of (3.14) with respect to $v$. Second, performing the Fourier transform and using Cauchy's theorem, we shift the contour of the integrals on the left hand side to the real axis. Finally performing the inverse Fourier transform and then integrating over $v$, we obtain the following NLIEs:

$$\log Y_j(v) = p_j(v) + g_j(v) + \sum_{l=1}^{\nu} K_{jl} * \log(1 + Y_l)[v] \quad (1 \le j \le \nu), \tag{4.4}$$

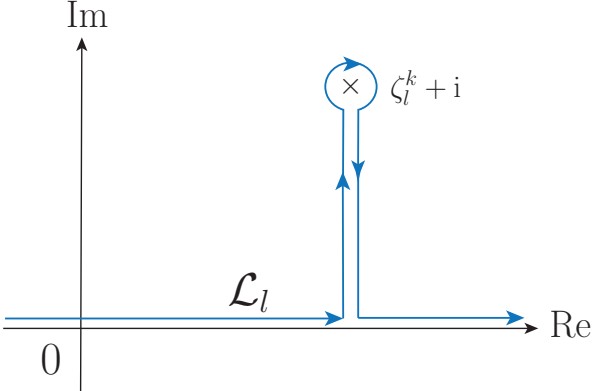

Figure 2: Depiction of a modified contour for $1 + Y_l(v)$ in (4.8). The contour is modified such that it encircles $\zeta_l^k + \mathrm{i}$ ($1 \le k \le m_l$) in clockwise direction.

where

$$K_{jk}(v) := A_{jk}s(v), \quad s(v) := \frac{1}{4\,\mathrm{ch}\,\frac{\pi}{2}v}, \tag{4.5}$$

and $A_{jk}$ denotes the adjacency matrix associated with the Dynkin diagram of type $D_\nu$ (see Fig. 1). The symbol $*$ in (4.4) denotes the convolution defined by $s * f(v) := \int_{-\infty}^{\infty} s(v - x)f(x)\mathrm{d}x$. The leading terms $p_j(v)$ and $g_j(v)$ in (4.4) are, respectively, given by

$$p_j(v) := (\delta_{j1} + \delta_{j2}\delta_{\nu 2})(Lp(v) + s'(v - \mathrm{i})\psi) - \delta_{j\nu-1}\frac{\nu}{2}\phi\mathrm{i} + \delta_{j\nu}\frac{\nu}{2}\phi\mathrm{i},$$

$$g_j(v) := \sum_{l=1}^{\nu} A_{jl} \sum_{k=1}^{m_l} p(v - \zeta_l^k), \tag{4.6}$$

where

$$p(v) := \log \mathrm{th}\,\frac{\pi}{4}v. \tag{4.7}$$

Here the artificial parameter $\psi$, which should be set to zero after all calculations, has been introduced in the leading term $p_j(v)$ for later convenience. The NLIEs (4.4) together with the subsidiary conditions (4.3) give the solutions to the $Y$-system (3.14).

Modifying the integration contours in the convolutions so that they surround the parameters $\zeta_l^k + \mathrm{i}$ ($1 \le k \le m_l$) in clockwise manner (see Fig. 2), we can reduce the NLIEs to simpler forms (see appendix A in detail):

$$\log Y_j(v) = p_j(v) + \sum_{l=1}^{\nu} K_{jl} \underset{\mathcal{L}_l}{*} \log(1 + Y_l)[v], \tag{4.8}$$

where $\mathcal{L}_l$'s ($l = 1, \ldots, \nu$) denote the modified contours. For the ground state, $\mathcal{L}_l$'s are just straight lines.

Applying the same procedures to (3.13) (see also appendix A for the derivation), and using

the identity (3.6), one obtains the energy eigenvalues for arbitrary excited states:

$$
E(\phi) = E_{\rm gs} + A \sum_{j=1}^{\delta_{\nu 2}+1} \left[ \sum_{k=1}^{m_j} s(\zeta_j^k) + \frac{1}{2\pi{\rm i}} s' * \log(1+Y_j)[-{\rm i}] \right]
$$

$$
= E_{\rm gs} + \frac{A}{2\pi{\rm i}} \sum_{j=1}^{\delta_{\nu 2}+1} s' \underset{\mathcal{L}_j}{*} \log(1+Y_j)[-{\rm i}], \tag{4.9}
$$

where $E_{\rm gs}$ is the ground state energy given by

$$
E_{\rm gs} = \frac{JL}{2}\Delta - AL \int_{-\infty}^{\infty} a(v)s(v){\rm d}v, \quad a(v) := \frac{\sin\gamma}{2\nu({\rm ch}\,\gamma v - \cos\gamma)}. \tag{4.10}
$$

All the information of the state is included in the set of the contours $\{\mathcal{L}_l\}$: each set $\{\mathcal{L}_l\}$ defines a different state.

Thus the curvature of the energy spectrum (see (2.11)) is given by

$$
L\frac{d^2E}{d\phi^2}\bigg|_{\phi=0} = \frac{AL}{2\pi{\rm i}} \sum_{j=1}^{\delta_{\nu 2}+1} s' \underset{\mathcal{L}_j}{*} \partial_\phi^2 \log(1+Y_j)[-{\rm i}]\bigg|_{\phi=0}. \tag{4.11}
$$

## 5 Finite temperatures

Each set of parameters $\{\zeta_j^k\}$ gives a specific energy level via (4.9). For $L \gg 1$, the distributions corresponding to equilibrium states are characterized by the condition of minimizing the free energy. The thermodynamic quantities such as the internal energy $\langle E \rangle$ and $\langle Ld^2E/{\rm d}\phi^2 \rangle$, which is the quantity we like to calculate, can be evaluated by inserting the characteristic distributions into (4.9) and (4.11).

Let us determine for $L \gg 1$ the distributions describing the equilibrium. The quantity $1/L(\zeta_j^{k+1} - \zeta_j^k)$ yields the density of the parameters $\zeta_j^k$. For all $j < \nu$ these densities will appear to have the meaning of hole densities of strings in the TBA approach, and for $j = \nu$ the distribution function has the meaning of the density function of odd-strings. We simply denote all such densities by $\rho_j^{\rm h}(v)$. For all $j$ these functions describe the distribution of $\zeta_j^k$

$$
\rho_j^{\rm h}(\zeta_j^k) = \lim_{L\to\infty} \frac{1}{L(\zeta_j^{k+1} - \zeta_j^k)} \quad (1 \le j \le \nu), \tag{5.1}
$$

On the other hand the quantity $\lim_{L\to\infty}(I_j^{k+1} - I_j^k)/(L(\zeta_j^{k+1} - \zeta_j^k))$ gives the total density of holes and "particles":

$$
\lim_{L\to\infty} \frac{I_j^{k+1} - I_j^k}{L(\zeta_j^{k+1} - \zeta_j^k)} = \rho_j^{\rm h}(\zeta_j^k) + \rho_j(\zeta_j^k), \tag{5.2}
$$

where $I_j^k$ denote the quantum numbers occurring in (4.3). The substitution of (4.3) and (4.4) into (5.2) yields

$$
\rho_j^{\rm h}(v) + \rho_j(v) = -\lim_{L\to\infty} \frac{1}{2\pi{\rm i}L} \partial_v \log Y_j(v+{\rm i}) = s_j(v) + \sum_{l=1}^{\nu} K_{jl} * \rho_l^{\rm h}[v], \tag{5.3}
$$

where

$$s_j(v) := (\delta_{j1} + \delta_{j2}\delta_{\nu2})s(v), \tag{5.4}$$

and we have set the parameter $\psi = 0$ in (4.6). By use of the density functions $\rho_j^{\mathrm{h}}(v)$ (5.1) and the energy spectrum (4.9), one finds that the energy eigenvalues per site $e$ can be written as

$$e = e_{\mathrm{gs}} + A \int_{-\infty}^{\infty} s(v) \left\{ \rho_1^{\mathrm{h}}(v) + \delta_{\nu2}\rho_\nu^{\mathrm{h}}(v) \right\} \mathrm{d}v, \tag{5.5}$$

where $e_{\mathrm{gs}} := \lim_{L\to\infty} E_{\mathrm{gs}}/L$ is the ground state energy per site (see (4.10)). The entropy per site $s$ is evaluated by taking the logarithm of the number of ways to take $\rho_j^{\mathrm{h}}(v)dv$ objects from a sequence of $(\rho_j^{\mathrm{h}}(v) + \rho_j(v))dv$ objects:

$$s = \sum_{j=1}^{\nu} \int_{-\infty}^{\infty} \left[ (\rho_j(v) + \rho_j^{\mathrm{h}}(v))\log(\rho_j(v) + \rho_j^{\mathrm{h}}(v))dv - \rho_j(v)\log\rho_j(v) - \rho_j^{\mathrm{h}}(v)\log\rho_j^{\mathrm{h}}(v) \right] \mathrm{d}v. \tag{5.6}$$

Minimizing the free energy, $\delta f = \delta(e - Ts) = 0$, we derive the so-called TBA equations determining the density functions $\rho_j^{\mathrm{h}}(v)$ $(1 \le j \le \nu)$ which characterize the equilibrium state:

$$\log \eta_j(v) = -\beta A s_j(v) + \sum_{l=1}^{\nu} K_{jl} * \log(1 + \eta_l)[v], \tag{5.7}$$

where we have defined

$$\eta_j(v) := \frac{\rho_j^{\mathrm{h}}(v)}{\rho_j(v)} \quad (1 \le j \le \nu). \tag{5.8}$$

This equation exactly agrees with the one derived by the string hypothesis [33, 34]. However, we emphasize that our formula does not rely on the string hypothesis, but only on the simple analytical assumption explained previously. In particular, the parameters $\zeta_j^k$ are real and hence no finite size dependent corrections of imaginary parts appear. The comparison of (5.7) with (5.3) leads to

$$\rho_j^{\mathrm{h}}(v) = -\frac{1}{A}\partial_\beta \log(1 + \eta_j(v)) \quad (1 \le j \le \nu). \tag{5.9}$$

Substituting these density functions into the NLIEs (4.4), we obtain the "thermal" NLIEs (not to be confused with the TBA equations):

$$\log Y_j^{\mathrm{th}}(v) = p_j(v) + g_j^{\mathrm{th}}(v) + \sum_{l=1}^{\nu} K_{jl} * \log(1 + Y_l^{\mathrm{th}})[v]$$

$$= p_j(v) + \sum_{l=1}^{\nu} K_{jl} \underset{\mathcal{L}_l}{*} \log(1 + Y_l^{\mathrm{th}})[v], \tag{5.10}$$

with

$$g_j^{\mathrm{th}}(v) = L \sum_{l=1}^{\nu} A_{jl}\, p * \rho_l^{\mathrm{h}}[v]. \tag{5.11}$$

Thus combining (5.10), (4.11) and (2.11), the Drude weight for finite temperature is given by

$$D = \lim_{L\to\infty} \left\langle L\frac{\mathrm{d}^2 E(\phi)}{\mathrm{d}\phi^2} \right\rangle = \lim_{L\to\infty} \frac{AL}{2\pi\mathrm{i}} \sum_{j=1}^{1+\delta_{\nu2}} s' \underset{\mathcal{L}_j}{*} \partial_\phi^2 \log(1 + Y_j^{\mathrm{th}})[-\mathrm{i}]\Big|_{\phi=0,\psi=0}. \tag{5.12}$$

# 6 Analysis of the Drude weight

In this section, we derive a manageable formula for the Drude weight for finite temperature by analyzing the thermal NLIEs (5.10) together with the TBA equations (5.7). We evaluate both temperature and size dependences of the Drude weight.

## 6.1 Analysis of the NLIEs

By using the techniques of the dressed function formalism as shown in appendix B, the Drude weight is rewritten as (B.8). Let us simplify this equation step by step. First, from (5.10) we find that

$$\partial_\phi \log Y_j^{\mathrm{th}}(v)\Big|_{\phi=0,\psi=0} = -\frac{\nu i}{2}\delta_{j\nu-1} + \frac{\nu i}{2}\delta_{j\nu}. \tag{6.1}$$

Furthermore, the relation $Y_\nu^{\mathrm{th}}(v) = Y_{\nu-1}^{\mathrm{th}}(v)$ for $\phi = 0$ reduces (B.8) to

$$D = -\frac{AL\nu^2}{4\pi i} \int_{\mathcal{L}_\nu} dv \frac{\partial_\psi \log(1 + Y_\nu^{\mathrm{th}}(v))}{1 + Y_\nu^{\mathrm{th}}(v)}\Bigg|_{\phi=0,\psi=0}. \tag{6.2}$$

In the following, as already done above, we carry out derivatives with respect to $\phi$ and/or $\psi$ and finally set these parameters to zero, which is sometimes done implicitly by omitting the symbols $\phi = 0$ and $\psi = 0$ for brevity. Modifying the integrand as

$$\frac{\partial_\psi \log(1 + Y)}{1 + Y} = \frac{Y \partial_\psi \log Y}{(1 + Y)^2} = -\frac{\partial_\psi \log Y}{\partial_v \log Y}\partial_v\left(\frac{1}{1 + Y}\right), \tag{6.3}$$

and integrating by parts we find

$$D = \frac{AL\nu^2}{4\pi i}\left[\frac{\partial_\psi \log Y_\nu^{\mathrm{th}}(v)}{\partial_v \log Y_\nu^{\mathrm{th}}(v)}\frac{1}{1 + Y_\nu^{\mathrm{th}}(v)}\Bigg|_{-\infty}^{\infty} - \int_{\mathcal{L}_\nu}\frac{dv}{1 + Y_\nu^{\mathrm{th}}(v)}\partial_v\left\{\frac{\partial_\psi \log Y_\nu^{\mathrm{th}}(v)}{\partial_v \log Y_\nu^{\mathrm{th}}(v)}\right\}\right]. \tag{6.4}$$

The factor $1/(1 + Y_\nu^{\mathrm{th}}(v))$ in the integrand of the above equation has poles at $v = \zeta_\nu^k + i$ with residues

$$\frac{1}{Y_\nu^{\mathrm{th}\prime}(\zeta_\nu^k + i)} = \frac{1}{Y_\nu^{\mathrm{th}}(\zeta_\nu^k + i)\log' Y_\nu^{\mathrm{th}}(\zeta_\nu^k + i)} = -\frac{1}{\log' Y_\nu^{\mathrm{th}}(\zeta_\nu^k + i)}. \tag{6.5}$$

Using Cauchy's theorem, we modify the integration contour $\mathcal{L}_\nu$ to the straight line (see Fig. 2). Then performing again an integration by parts, which cancels the surface terms, we arrive at

$$\begin{aligned}
D &= -\frac{AL\nu^2}{2}\sum_{k=1}^{m_\nu}\frac{1}{\partial_v \log Y_\nu^{\mathrm{th}}(v)}\partial_v\left\{\frac{\partial_\psi \log Y_\nu^{\mathrm{th}}(v)}{\partial_v \log Y_\nu^{\mathrm{th}}(v)}\right\}\Bigg|_{v=\zeta_\nu^k+i} \\
&\quad + \frac{AL\nu^2}{4\pi i}\int_{-\infty}^{\infty}dv\partial_v\left\{\frac{1}{1 + Y_\nu^{\mathrm{th}}(v)}\right\}\frac{\partial_\psi \log Y_\nu^{\mathrm{th}}(v)}{\partial_v \log Y_\nu^{\mathrm{th}}(v)} \\
&= -\frac{AL^2\nu^2}{2}\int_{-\infty}^{\infty}dv\frac{\rho_\nu^{\mathrm{h}}(v)}{\partial_v \log Y_\nu^{\mathrm{th}}(v+i)}\partial_v\left\{\frac{\partial_\psi \log Y_\nu^{\mathrm{th}}(v+i)}{\partial_v \log Y_\nu^{\mathrm{th}}(v+i)}\right\} \\
&\quad + \frac{AL\nu^2}{4\pi i}\int_{-\infty}^{\infty}dv\partial_v\left\{\frac{1}{1 + Y_\nu^{\mathrm{th}}(v)}\right\}\frac{\partial_\psi \log Y_\nu^{\mathrm{th}}(v)}{\partial_v \log Y_\nu^{\mathrm{th}}(v)}.
\end{aligned} \tag{6.6}$$

Note that we have implicitly taken $\phi \to 0$ and $\psi \to 0$ as explained above.

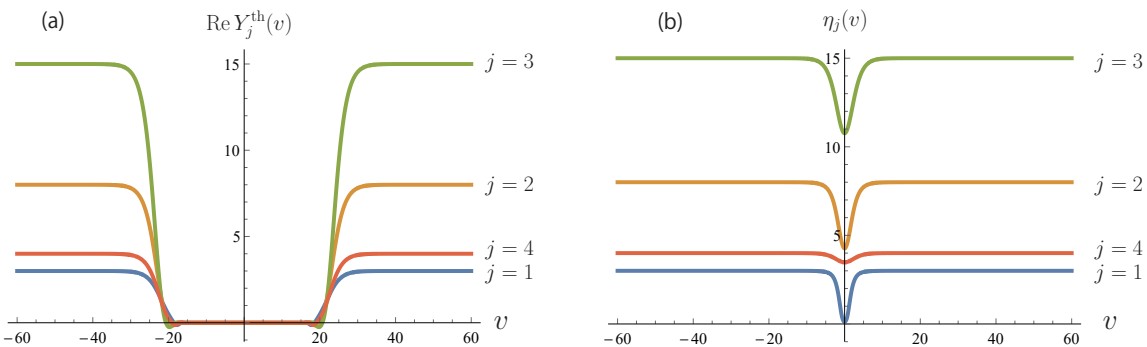

Figure 3: The behavior of the functions $Y_j^{\text{th}}(v)$ and $\eta_j(v)$ satisfying (5.10) and (5.7) for $\nu = 5$, $\beta J = 1$ and $L = 10^6$. The functions $Y_j^{\text{th}}(v)$ show a sharp crossover at about the points $v = \pm v_L = \pm(\nu/\pi)\log L \simeq 22$. On the other hand, the function $\eta_j(v)$ smoothly vary in a small region.

Let us consider the thermodynamic limit $L \to \infty$. For $L \gg 1$, by some simple analysis of the thermal NLIEs (5.10), one observes that the $Y$-functions $Y_j^{\text{th}}(v)$ show a sharp crossover at about the points $v = \pm v_L := \pm(\nu/\pi)\log L$:

$$Y_j^{\text{th}}(v) \simeq \begin{cases} j(j+2) & (|v| > v_L) \\ 0 & (|v| < v_L) \end{cases} (j < \nu - 1), \ Y_{\nu-1}^{\text{th}}(v) = Y_\nu^{\text{th}}(v) = \begin{cases} \nu - 1 & (|v| > v_L) \\ 0 & (|v| < v_L) \end{cases}.$$

See Fig. 3 (a) for the behavior of $Y_j^{\text{th}}(v)$ for $\nu = 5$, $\beta J = 1$ and $L = 10^6$. On the other hand, for arbitrary finite temperatures the behavior of the functions $\eta_j(v)$ is much smoother than that of $Y_j^{\text{th}}(v)$. See Fig. 3 (b) for the behavior of $\eta_j(v)$ for $\nu = 5$ and $\beta J = 1$. Generally, $\eta_j(v)$ smoothly vary in a narrow region, typically $|v| < v_\beta$ where $v_\beta \simeq (\nu/\pi)\log(1 + \beta) \ll v_L$ and become constant outside this region.

$$\eta_j(v) \simeq \begin{cases} j(j+2) & (j < \nu - 1) \\ \nu - 1 & (j = \nu - 1, \ j = \nu) \end{cases} \quad (|v| > v_\beta). \tag{6.7}$$

Thus to analyze (6.6), one must carefully take into account this behavior of $Y_j^{\text{th}}(v)$ and $\eta_j(v)$. First we consider $\partial_v \log Y_j^{\text{th}}(v)$. Thanks to (5.7), (5.9) and (5.10), we identify

$$\partial_v \log Y_j^{\text{th}}(v + \mathrm{i}) = \frac{2\pi\mathrm{i}L}{A}\partial_\beta \log \eta_j(v) + \sum_{l=1}^{\nu} K_{jl} * \frac{\partial_v \log Y_l^{\text{th}}}{1 + 1/Y_l^{\text{th}}}[v + \mathrm{i}]. \tag{6.8}$$

Due to the crossover behavior, the second term on the right hand side of the above equation does not contribute for $|v| < v_L$, and therefore

$$\partial_v \log Y_j^{\text{th}}(v + \mathrm{i}) \simeq \frac{2\pi\mathrm{i}L}{A}\partial_\beta \log \eta_j(v) \quad \text{for } |v| < v_L. \tag{6.9}$$

Though both functions $\partial_v \log Y_j^{\text{th}}(v)$ and $(2\pi\mathrm{i}L/A)\partial_\beta \log \eta_j(v)$ exponentially converge to zero as $v \to \pm\infty$, the decay rates are, in general, different due to the $O(1)$ contribution of the second term in (6.8) for $|v| > v_L$.

Next we consider $\partial_\psi \log Y_j^{\text{th}}(v)$. Using the same technique to derive (6.6), we have

$$\partial_\psi \log Y_j^{\text{th}}(v+\mathrm{i}) = s_j'(v) - 2\pi\mathrm{i}L \sum_{l=1}^{\nu} \int_{-\infty}^{\infty} \mathrm{d}x K_{jl}(v-x) \frac{\partial_\psi \log Y_l^{\text{th}}(x+\mathrm{i})}{\partial_x \log Y_l^{\text{th}}(x+\mathrm{i})} \rho_l^{\text{h}}(x)$$
$$+ \sum_{l=1}^{\nu} K_{jl} * \frac{\partial_\psi \log Y_\nu^{\text{th}}}{1+1/Y_l^{\text{th}}}[v+\mathrm{i}]. \tag{6.10}$$

Inside the region $|v| < v_L$, the same argument to obtain (6.8) is also applicable: we can ignore the third term on the right hand side of the above equation. The insertion of (5.9) and (6.9) into the above leads to

$$\partial_\psi \log Y_j^{\text{th}}(v+\mathrm{i}) = s_j'(v) + \sum_{l=1}^{\nu} \int_{-\infty}^{\infty} \mathrm{d}x\, K_{jl}(v-x) \frac{\partial_\psi \log Y_l^{\text{th}}(x+\mathrm{i})}{1+1/\eta_l(x)}. \tag{6.11}$$

Comparing this equation with the equation derived by taking the derivative of (5.7) with respect to $v$, we find

$$\partial_\psi \log Y_j^{\text{th}}(v+\mathrm{i}) = -\frac{1}{\beta A} \partial_v \log \eta_j(v) \quad \text{for } |v| < v_L. \tag{6.12}$$

To describe $\partial_\psi \log Y_j^{\text{th}}(v)$ in the whole region, we must consider the correction term:

$$\partial_\psi \log Y_j^{\text{th}}(v+\mathrm{i}) = -\frac{1}{\beta A} \partial_v \log \eta_j(v) + \sum_{l=1}^{\nu} K_{jl} * \frac{\partial_\psi \log Y_l^{\text{th}}}{1+1/Y_l^{\text{th}}}[v+\mathrm{i}]. \tag{6.13}$$

Again, both quantities $\partial_\psi \log Y_j^{\text{th}}(v)$ and $-(1/\beta A)\partial_v \log \eta_j(v)$ exponentially converge to zero as $v \to \pm\infty$ with different exponents. For the ratios, however, we find the relation

$$\frac{\partial_\psi \log Y_j^{\text{th}}(v+\mathrm{i})}{\partial_v \log Y_j^{\text{th}}(v+\mathrm{i})} \simeq -\frac{1}{2\pi\mathrm{i}L\beta} \frac{\partial_v \log \eta_j(v)}{\partial_\beta \log \eta_j(v)} \tag{6.14}$$

holds in the whole region, which follows from (6.9) and (6.12) together with the fact that the driving terms of the linear integral equations (6.8) and (6.13) converge exponentially to zero with the same exponent for $|v| > v_\beta (\ll v_L)$.

Now we evaluate the Drude weight (6.6) in the thermodynamic limit. As for the first term in the second equation of (6.6) (let us denote it as $D_1$), the ratio of derivatives of $\log Y_\nu$ can be replaced by those of $\log \eta_\nu$:

$$D_1 = \frac{AL\nu^2}{4\pi\mathrm{i}\beta} \int_{-\infty}^{\infty} \mathrm{d}v \frac{\rho_\nu^{\text{h}}(v)}{\partial_v \log Y_\nu^{\text{th}}(v+\mathrm{i})} \partial_v \left\{ \frac{\partial_v \log \eta_\nu(v)}{\partial_\beta \log \eta_\nu(v)} \right\}. \tag{6.15}$$

Since the second factor in the above integrand converges rapidly to zero for $|v| > v_\beta (\ll v_L)$, one can neglect the behavior of the first factor around $v = \pm v_L$. Consequently, we can replace the denominator of the first factor by use of (6.9):

$$D_1 = \frac{A}{8\gamma^2\beta} \int_{-\infty}^{\infty} \mathrm{d}v \frac{\eta_\nu(v)}{1+\eta_\nu(v)} \partial_v \left\{ \frac{\partial_v \log \eta_\nu(v)}{\partial_\beta \log \eta_\nu(v)} \right\}$$
$$= \frac{A}{4\gamma^2\beta} \frac{\nu-1}{\nu} \frac{\partial_v \log \eta_\nu(\infty)}{\partial_\beta \log \eta_\nu(\infty)} - \frac{A}{8\gamma^2\beta} \int_{-\infty}^{\infty} \mathrm{d}v \frac{\eta_\nu(v)\{\partial_v \log \eta_\nu(v)\}^2}{\{1+\eta_\nu(v)\}^2 \partial_\beta \log \eta_\nu(v)}, \tag{6.16}$$

where we have also substituted (5.9) and the asymptotic value (6.7). For the second term in (6.6) (denoting it as $D_2$), we notice that the first factor in the integrand is zero except for a small region around $v = \pm v_L$. Hence the second factor can be replaced by a ratio of derivatives of $\log \eta$ functions. This ratio is in fact constant for $|v| > v_\beta (\ll v_L)$, hence we obtain

$$D_2 = \frac{-A}{4\gamma^2\beta} \frac{\partial_v \log \eta_\nu(\infty)}{\partial_\beta \log \eta_\nu(\infty)} \int_0^\infty \mathrm{d}v \, \partial_v \left( \frac{1}{1 + 1/Y_\nu^{\mathrm{th}}(v)} \right) = \frac{-A}{4\gamma^2\beta} \frac{\nu - 1}{\nu} \frac{\partial_v \log \eta_\nu(\infty)}{\partial_\beta \log \eta_\nu(\infty)}, \qquad (6.17)$$

where we have inserted (6.7). In consequence, the surface term in (6.16) is exactly cancelled by the second term $D_2$, and then

$$D = -\frac{A}{8\gamma^2\beta} \int_{-\infty}^\infty \mathrm{d}v \, \frac{\eta_\nu(v) \left\{ \partial_v \log \eta_\nu(v) \right\}^2}{\left\{ 1 + \eta_\nu(v) \right\}^2 \partial_\beta \log \eta_\nu(v)}. \qquad (6.18)$$

This expression exactly coincides with the Drude weight derived by Zotos [5]. Especially for the $XY$ (free fermion) model ($\nu = 2$), all the functions $Y_j^{\mathrm{th}}(v)$ and $\eta_j(v)$ are explicitly given by $\log Y_j^{\mathrm{th}}(v) = p_j(v)$ and $\log \eta_j(v) = -\beta s_j(v)$. Then the Drude weight reads

$$D(\nu = 2) = \frac{\beta J^2}{\pi} \int_{-\pi/2}^{\pi/2} \mathrm{d}p \, \frac{\sin^2 p}{\mathrm{ch}^2(\beta J \cos p)}. \qquad (6.19)$$

This quantity is nothing but $\langle -K \rangle / L$ as expected in (2.8). (Note that the spin current $\mathcal{J}_{\mathrm{s}}$ (2.5) is a conserved quantity for the $XY$ case.)

## 6.2   Numerical evaluation

As shown in the previous subsection, the Drude weight $D$ for arbitrary system size $L$ is given by (6.6), and converges in the thermodynamic limit $L \to \infty$ to the result derived by Zotos (6.18). Here we evaluate (6.6) numerically and examine how this quantity converges to the results in the limit $L \to \infty$. This can be achieved by numerically solving the NLIEs (5.10), (6.8) and (6.10) together with the TBA equations (5.7) and (5.9) which determine the hole densities $\rho_j^{\mathrm{h}}(v)$ for the equilibrium state. For finite size $L$ we use discrete distributions that approximate the continuous densities as closely as possible.

In Fig. 4, the large-size asymptotic behavior of the Drude weight $D$ is shown for various interaction strengths $\Delta = \cos \pi/\nu$ ($2 \leq \nu \leq 6$) with fixed temperature $\beta J = 1$. We observe that $D$ sensitively depends on the system size (or equivalently on the number of holes) even for $L \simeq 10^3$. In contrast to this, for the $XY$ (free fermion) model ($\nu = 2$), where the Drude weight in the thermodynamic limit is simply given by the bulk quantities (6.19), $D$ does not exhibit such a strong size dependence. In general, the bulk quantities such as the internal energy quickly converge to the results for the thermodynamic limit (5.5).

The size-dependence of the Drude weight $D$ becomes more prominent with increase of the interaction strength. However, $D$ eventually converges to Zotos' results which are separately shown in Fig. 6. Note that the Drude weight $D$ converges to zero in the isotropic limit $\Delta = 1$ [5]. The dependence on the system size is more significant with decrease of the temperature (or equivalently, with the decrease of the number of holes), which follows from Fig 5. This strong size-dependence may explain that extrapolations from various numerical methods such as numerical diagonalizations and quantum Monte Carlo methods present contradictory results.

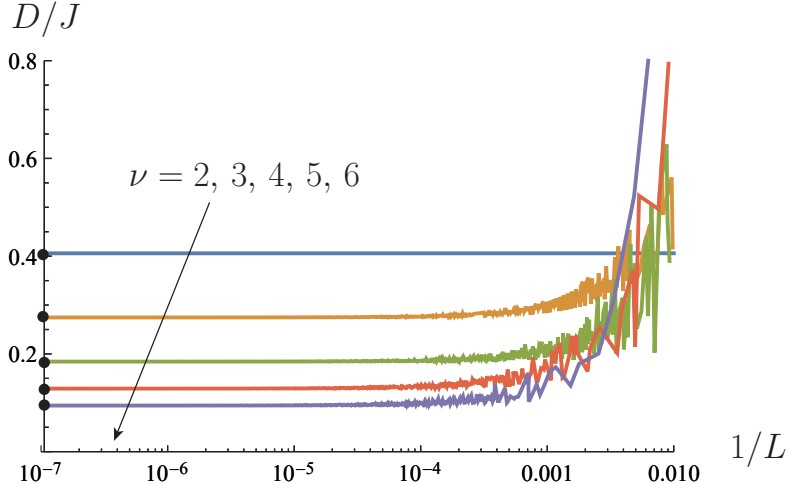

Figure 4: The large-size asymptotics of the Drude weight $D$ (6.6) for $2 \leq \nu \leq 6$ and fixed temperature $\beta J = 1$. The Drude weight strongly depends on the system size even for large system size $L \simeq 10^3$, and quite slowly converges to the results derived by Zotos [5] (see also (6.18)) which are depicted by filled circles. In other words, the Drude weight sensitively depends on the number of holes especially for small $L$. In contrast, for the $XY$ chain ($\nu = 2$), $D$ does not show such a strong size dependence. The size dependence becomes more significant with the increase of $\Delta = \cos \pi/\nu$.

## 7    Summary and Discussions

In this paper, we have analyzed the spin transport properties of the spin-1/2 Heisenberg $XXZ$ chain for finite temperature. We have constructed the $T$- and $Y$-systems which are functional relations satisfied by the row-to-row transfer matrices and certain combinations thereof ($Y$-functions). This enabled us to convert the problem of analyzing the energy spectrum into the problem of examining the analytical properties of the $Y$-functions. Consequently, the energy eigenvalues for arbitrary excited states can be written as the solutions to non-linear integral equations (NLIEs) satisfied by the $Y$-functions. Combining this with the thermodynamic Bethe ansatz (TBA) which determines the equilibrium state, we have succeeded in describing the Drude weight in terms of specific $Y$-functions compatible with the solutions to the TBA equations. Analyzing the NLIEs and TBA equations, we have evaluated the large-size asymptotic behavior of the Drude weight. We found that the Drude weight sensitively depends on the system size, i.e., the number of holes, and shows very slow convergence in the thermodynamic limit $L \to \infty$ to the results obtained by Zotos [5]. This sensitive and strong finite-size dependence might explain the difficulties to evaluate the Drude weight by extrapolations of numerical approaches.

There exist several natural extensions of our method. The first one is an extension to the case of general rational numbers $\nu = p/q$ where $p$ and $q$ are positive coprime integers satisfying $p \geq 2q$. The $Y$-system corresponding to (3.14) should be modified along the lines of [30]. We expect that the resultant formula of the Drude weight in the thermodynamic limit will recover that derived by a different approach [24]. In fact, the Drude weight at finite temperatures is

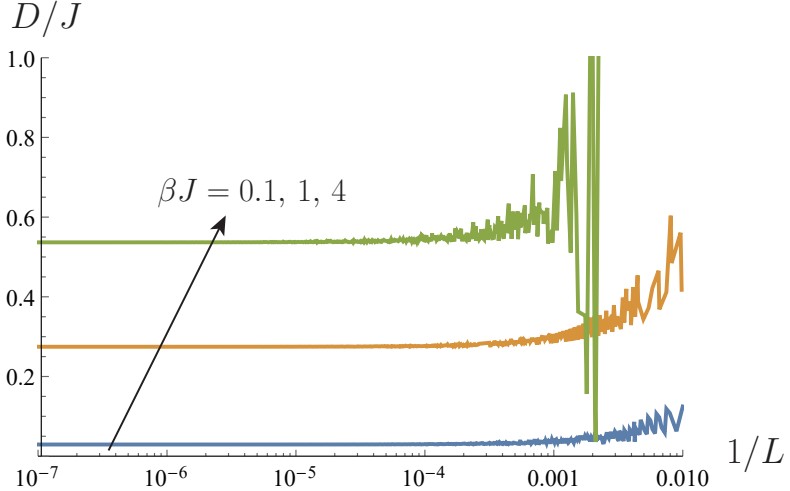

Figure 5: The large size-asymptotics of the Drude weight $D$ for $\nu = 3$ and different temperatures $\beta J = 0.1, 1, 4$. With decrease of the temperature, the size dependence becomes more prominent.

considered to be an everywhere discontinuous function of the anisotropy parameter $\Delta$: the Drude weight exhibits a fractal dependence on $\Delta$ [18, 19, 24]. For more quantitative and rigorous analysis of this intriguing behavior, a formula describing the Drude weight for any irrational numbers $\nu$ is highly desired.

Another simple extension is to consider the model with external magnetic fields. This can be done by replacing the driving term $s_j(v)$ (5.4) of the TBA equations (5.7) by

$$s_j(v) = (\delta_{j1} + \delta_{j2}\delta_{\nu 2})s(v) + \frac{\nu}{2}\beta h(\delta_{j\nu-1} - \delta_{j\nu}). \tag{7.1}$$

In this case, the thermomagnetic effects such as the spin Seebeck effect play a crucial role in the transport properties [35–37].

Finally, the effect of boundary conditions is also an interesting problem. For the twisted boundary conditions with a twist angle $\alpha \in \mathbb{R}$, we can derive a formula by just replacing $\phi \to \phi + \alpha$ in our formulation. The analytic treatments, however, are more complicated, since all the summands in (B.8) contribute to the Drude weight, in contrast to $\alpha = 0$ where only two summands $j = \nu - 1$, $j = \nu$ survive.

## Acknowledgements

The present work was partially supported by Grant-in-Aid for Scientific Research (C) No. 16K05468 from Japan Society for the Promotion of Science and by research unit FOR 2316 of DFG. AK acknowledges interesting and helpful discussions with X. Zotos.

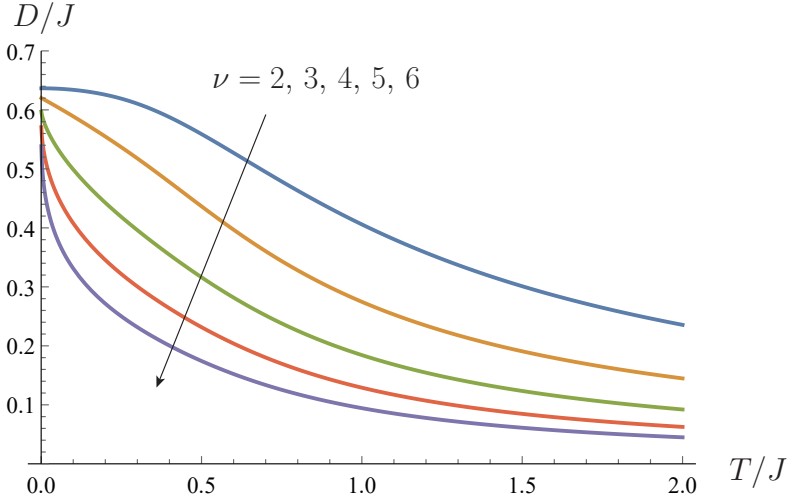

Figure 6: The temperature dependence of the Drude weight $D$ for $2 \leq \nu \leq 6$ and $L = \infty$.

## A    Derivation of NLIEs

Let us explain how to derive the NLIEs in section 4. Here we restrict ourselves to the case $\nu > 3$. A similar treatment is also possible for $\nu \leq 3$.

In general, we solve the functional equations in multiplicative form by taking the logarithmic derivative and apply the Fourier transform. In order to have a Fourier representation with convergence in a sufficiently wide strip in the complex plane we have to render the functions analytic in such a strip. This is achieved by suitable "counter terms". For instance, the NLIE for $j = 1$ in (4.4) can be derived in the following manner. Due to (3.12), one finds that, in the physical strip $\mathrm{Im}\, v \in [-1, 1]$, the $Y$-function $Y_1(v)$ has zeros at $v = \zeta_2^k$ $(k = 1, \ldots, m_2)$ and a zero of order $L$ at $v = 0$. Define a modified function as

$$\widetilde{Y}_1(v) = \frac{Y_1(v)}{(\mathrm{th}\, \frac{\pi}{4} v)^L \prod_{k=1}^{m_2} \mathrm{th}\, \frac{\pi}{4} \left(v - \zeta_2^k\right)}. \tag{A.1}$$

(Note that the second factor in the denominator is not needed for the $XY$ case ($\nu = 2$), and one more additional factor $\prod_{k=1}^{m_3} \mathrm{th}\, \frac{\pi}{4}(v - \zeta_3^k)$ is needed for $\nu = 3$). Utilizing the identity $\mathrm{th}\, \frac{\pi}{4}(v + \mathrm{i})\, \mathrm{th}\, \frac{\pi}{4}(v - \mathrm{i}) = 1$, we rewrite the $Y$-system (3.14) as

$$\widetilde{Y}_1(v + \mathrm{i})\widetilde{Y}_1(v - \mathrm{i}) = 1 + Y_2(v). \tag{A.2}$$

The functions $\widetilde{Y}_1(v)$ and $1 + Y_2(v)$ (see (3.13)) are analytic and nonzero in the physical strip, and have constant asymptotics. Taking the logarithmic derivative of both sides of (A.2), applying the Fourier transform

$$\widehat{f}(k) = \mathcal{F}[f(v)] := \int_{\infty}^{\infty} f(v) \mathrm{e}^{\mathrm{i}kv} \mathrm{d}v, \tag{A.3}$$

and then shifting the integration contours of the left hand side of the resultant equation, we have

$$2 \, \mathrm{ch}\, k \, \mathcal{F}[\partial_v \log \widetilde{Y}_1(v)] = \mathcal{F}[\partial_v \log(1 + Y_2(v))]. \tag{A.4}$$

Performing the inverse Fourier transform

$$f(v) = \mathcal{F}^{-1}[\widehat{f}(k)] := \frac{1}{2\pi} \int_{-\infty}^{\infty} \widehat{f}(k)\mathrm{e}^{-\mathrm{i}kv}\mathrm{d}v, \tag{A.5}$$

and using the convolution integral

$$\mathcal{F}^{-1}[f(k)g(k)] = f * g(v) := \int_{-\infty}^{\infty} f(v-x)g(x)\mathrm{d}x, \tag{A.6}$$

we finally arrive at

$$\log Y_1(v) = L \log \operatorname{th} \frac{\pi}{4}v + \sum_{k=1}^{m_2} \log \operatorname{th} \frac{\pi}{4}(v - \zeta_2^k) + s * \log(1 + Y_2)(v), \tag{A.7}$$

where

$$s(v) := \mathcal{F}^{-1}\left[\frac{1}{2\operatorname{ch}k}\right] = \frac{1}{4\operatorname{ch}\frac{\pi}{2}v}. \tag{A.8}$$

The function $1 + Y_2(v)$ appearing in the convolution in (A.7) has zeros $\zeta_2^k + \mathrm{i}$ ($1 \le k \le m_2$) which follows by (3.13), and hence we have

$$
\begin{aligned}
\int_{\mathcal{L}_2} s'(v-x)\log(1 + Y_2(x))\mathrm{d}x &= \int_{\mathcal{L}_2} s(v-x)\partial_x \log(1 + Y_2(x))\mathrm{d}x \\
&= -\sum_{k=1}^{m_2} \oint_{\zeta_2^k+\mathrm{i}} s(v-x)\partial_x \log\left\{\operatorname{sh}\frac{\gamma}{2}(x - \zeta_2^k - \mathrm{i})\right\} + s' * \log(1 + Y_2)(v) \\
&= -2\pi\mathrm{i}\sum_{k=1}^{m_2} s(v - \zeta_2^k - \mathrm{i}) + s' * \log(1 + Y_2)(v) \\
&= \partial_v \log \operatorname{th}\frac{\pi}{4}(v - \zeta_2^k) + s' * \log(1 + Y_2)(v),
\end{aligned} \tag{A.9}
$$

where $\mathcal{L}_2$ is the contour modified such that it encircles $\zeta_2^k + \mathrm{i}$ ($1 \le k \le m_2$) (see Fig. 2 in section 4). The combination of this with (A.7) yields

$$\log Y_1(v) = L \log \operatorname{th}\frac{\pi}{4}v + s \underset{\mathcal{L}_2}{*} \log(1 + Y_2)(v). \tag{A.10}$$

All the other NLIEs in (4.4) or (4.8) can be derived in completely the same way.

Finally let us derive the energy eigenvalue (4.9) from the first relation in (3.13):

$$T_1(v + \mathrm{i})T_1(v - \mathrm{i}) = \varphi(v + 2\mathrm{i})\varphi(v - 2\mathrm{i})(1 + Y_1(v)). \tag{A.11}$$

Let us define

$$\widetilde{T}_1(v) = \frac{T_1(v)}{\varphi(v - \mathrm{i}) \prod_{k=1}^{m_1} \operatorname{th}\frac{\pi}{4}(v - \zeta_1^k)}. \tag{A.12}$$

(Note that, for $\nu = 2$, one more additional factor $\prod_{k=1}^{m_2} \operatorname{th}\frac{\pi}{4}(v - \zeta_2^k)$ is needed in the denominator.). Then one finds that $\widetilde{T}_1(v)$ is analytic and nonzero in $\operatorname{Im}v \in [-1 - \epsilon_1, 1 - \epsilon_2]$ where $\epsilon_1$ and $\epsilon_2$ are some small positive real numbers, and also $\widetilde{T}_1(v)$ has constant asymptotics. Eq. (A.11) is then modified as

$$\widetilde{T}_1(v + \mathrm{i})\widetilde{T}_1(v - \mathrm{i}) = \frac{\varphi(v + 2\mathrm{i})}{\varphi(v)}(1 + Y_1(v)). \tag{A.13}$$

Applying a procedure similar to the derivation of the NLIEs (A.7) and (A.10)

$$\partial_v \log T_1(v - \mathrm{i}) = \partial_v \log \varphi(v - 2\mathrm{i}) - 2\pi\mathrm{i}Ls * a(v) + 2\pi\mathrm{i} \sum_{k=1}^{m_1} s(v - \zeta_1^k) + s' * \log(1 + Y_1)(v - \mathrm{i})$$

$$= \partial_v \log \varphi(v - 2\mathrm{i}) - 2\pi\mathrm{i}Ls * a(v) + s' \underset{\mathcal{L}_1}{*} \log(1 + Y_1)(v - \mathrm{i}). \tag{A.14}$$

The application of the formula (3.6) into the above equation leads to (4.9).

## B Dressed function formalism

Here we briefly summarize a technique to analyze the Drude weight $D$ given by (5.12). Let us consider the logarithmic derivative of $Y$-functions:

$$\partial \log(1 + Y) = \frac{\partial \log Y}{1 + 1/Y},$$

$$\partial^2 \log(1 + Y) = \frac{\partial^2 \log Y}{1 + 1/Y} + \frac{(\partial \log Y)^2}{(1 + 1/Y)(1 + Y)}, \tag{B.1}$$

where $\partial$ denotes the differential operator with respect to an arbitrary parameter. Applying this relations to (4.8) (or (5.10)), one obtains

$$(1 + 1/Y_j)A_j = a_j + \sum_{l=1}^{\nu} K_{jl} \underset{\mathcal{L}_l}{*} A_l,$$

$$(1 + 1/Y_j)B_j = b_j + \sum_{l=1}^{\nu} K_{jl} \underset{\mathcal{L}_l}{*} B_l, \tag{B.2}$$

where

$$A_j := \partial \log(1 + Y_j), \qquad a_j := \partial p_j,$$

$$B_j := \partial^2 \log(1 + Y_j), \qquad b_j := \partial^2 p_j + \frac{(\partial \log Y)^2}{1 + Y}. \tag{B.3}$$

Multiplying the first (resp. second) equation in (B.2) by $B_j$ (resp. $A_j$), integrating over $v$ along the line $\mathcal{L}_j$ and summing the resultant equation over $j$, we find the following is valid:

$$\sum_{j=1}^{\nu} \int_{\mathcal{L}_j} \mathrm{d}v \left\{ \left(1 + \frac{1}{Y_j(v)}\right) A_j(v) - a_j(v) \right\} B_j(v)$$

$$= \sum_{j=1}^{\nu} \int_{\mathcal{L}_j} \mathrm{d}v \left\{ \left(1 + \frac{1}{Y_j(v)}\right) B_j(v) - b_j(v) \right\} A_j(v). \tag{B.4}$$

Here we have used the relation $K_{jl}(v) (= K_{lj}(v)) = K_{lj}(-v)$. Thus we obtain

$$\sum_{j=1}^{\nu} \int_{\mathcal{L}_j} \mathrm{d}v \, a_j(v) B_j(v) = \sum_{j=1}^{\nu} \int_{\mathcal{L}_j} \mathrm{d}v \, b_j(v) A_j(v). \tag{B.5}$$

Setting $A_j = \partial_\psi \log(1 + Y_j^{\text{th}})$ and $B_j(v) = \partial_\phi^2 \log(1 + Y_j^{\text{th}})$ in (5.10), and utilizing the above relation, we have

$$\sum_{j=1}^{\nu} \int_{\mathcal{L}_j} \mathrm{d}v \, \partial_\psi p_j(v) \partial_\phi^2 \log(1 + Y_j^{\text{th}}(v))$$
$$= \sum_{j=1}^{\nu} \int_{\mathcal{L}_j} \mathrm{d}v \left\{ \partial_\phi^2 p_j(v) + \frac{(\partial_\phi \log Y_j^{\text{th}}(v))^2}{1 + Y_j^{\text{th}}(v)} \right\} \partial_\psi \log(1 + Y_j^{\text{th}}(v)). \quad (\text{B.6})$$

The insertion of the relations $\partial_\psi p_j(v) = s'(v-i)\delta_{j1}$, $\partial_\phi^2 p_j(v) = 0$ yields

$$\sum_{j=1}^{1+\delta_{\nu 2}} \int_{\mathcal{L}_j} \mathrm{d}v s'(v-\mathrm{i}) \partial_\phi^2 \log(1 + Y_j^{\text{th}}(v))$$
$$= \sum_{j=1}^{\nu} \int_{\mathcal{L}_j} \mathrm{d}v \frac{(\partial_\phi \log Y_j^{\text{th}}(v))^2 \partial_\psi \log(1 + Y_j^{\text{th}}(v))}{1 + Y_j^{\text{th}}(v)}. \quad (\text{B.7})$$

As a result, the Drude weight (5.12) is expressed as

$$D = \lim_{L \to \infty} \frac{AL}{2\pi\mathrm{i}} \sum_{j=1}^{\nu} \int_{\mathcal{L}_j} \mathrm{d}v \frac{(\partial_\phi \log Y_j^{\text{th}}(v))^2 \partial_\psi \log(1 + Y_j^{\text{th}}(v))}{1 + Y_j^{\text{th}}(v)} \Bigg|_{\phi=0, \psi=0}. \quad (\text{B.8})$$

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
