# Peer review of "The spin Drude weight of the spin-1/2 $XXZ$ chain: An analytic finite size study"

_SciPost Physics_

## Round 1 · Referee Report · Anonymous · 2019-5-21

Strengths

1- New non-trivial results on the finite-size scaling of a thermodynamic quantity.

Weaknesses

1- The presentation is rather technical and unlikely suitable to non-experts.
2- Some important details appear to be missing.

Report

[1]

- In spite of some significant progress on the subject over the past few years, the introductory paragraphs, in particular the 2nd and 3rd paragraphs which largely mention 20 years old results, do not seem to be particularly appreciative of this. For example, "Nevertheless, by utilizing the integrability, several transport coeffcients of the model have been calculated within linear response theory.", is an obvious misrepresentation of the present understanding of quantum transport properties in integrable quantum systems, including the Heisenberg model considered here.

I would like to draw the authors attention to refs. [22,28] and additionally to [Ilievski and De Nardis, PRB 96(8), p.081118 (2017)]. There a powerful framework of hydrodynamics, developed in [25,26], has been used to obtain general and universal (that is state and quantity independent) closed-form expressions for the Drude weights valid for a broad range of integrable systems. Furthermore, there are even several exact results regarding spin transport on sub-balistic scales. I believe that, mostly for the reader's benefit, these developments deserve to be acknowledged in the introductory paragraphs in some transparent manner.

- I do not think referring to the "optimal lower bound" is an optimal choice of words. In fact, the bound of ref. [19] is now known to be the exact value of the spin Drude weight, see ref. [22]. In the infinite-temperature limit, the expression of [19] has been re-obtained explicitly in [Collura et al., PRB 97(8) p.081111 (2018)].

In the same paragaph: while [22,27] seem to be relevant references, [25,26,29] do not concern the Drude weights.

- The meaning of the so-called "extended" Thermodynamic Bethe Ansatz is a bit obscure I must say; the Drude weights are an equilibrium property after all, cf. [22,28]. Let me also mention [Ilievski and De Nardis, PRB 96(8), p.081118 (2017)] which equivalences between various approaches (i.e. the hydrodynamic formulae, Mazur equality, and Kohn's twisting formula employed by Zotos).

[2]

- Given that formula (11) is a well-known result, it may be worthwhile citing also ref. [5] and e.g. [Castella et al., PRL 74, 972 (1995)].

[3]

- "Instead of solving the BAE (3.8) directly, we introduce for convenience a more general family of transfer matrices (T-functions)..."

A small remark just to prevent confusion: the T-functions do not refer to transfer matrices but instead to their eigenvalues.

[4]

- I think the definition of $s^{\prime}$ (cf. Eq.(4.6)) is missing.
- The introduction of quantity $\psi$ in Eq. (4.6) seems rather ad-hoc at this point.
Frankly, even afterwards in Sec. 6.1. where $\psi$ appears to have an important role, its purpose is not particularly transparent. Hopefully the authors can expand on this.

- Missing hyperlink to the reference to Appendix A.

[5]

- Parameters $\zeta^k_j$, as introduced in Eq.(4.1), take in general complex values. But apparently here the so-called holes rapidities $\zeta^k_j$ are assumed to lie on the real axis.
Is this an assumption or it instead follows from some formal property of the T-system?

- "This equation exactly agrees with the one derived by the string hypothesis [33, 34]. However, we emphasize that our formula does not rely on the string hypothesis, but only on the simple analytical assumption explained previously."

Well, doesn't this suggest that the analytical assumption is simply equivalent to the string "hypothesis"?

- How to deal with the modified integration contour, defined in Eq.(4.8), in the thermodynamic limit when $\zeta^k_j$ condense in some way on the real axis. Here, in particular, I mean the second equality in Eq. (5.10) which looks ambiguous to me. Since in my view this is the key step in the context of finite-volume analysis, I hope the authors can give a precise explanation here.

- On a related note, in Eq.(5.9) the $L\to \infty$ limit has already been taken, while afterwards Eq. (5.12) explicitly depends on system size L. How should one therefore interpret the contour prescription?

- I noticed that Eq.(5.9), which eventually leads to Eq. (5.12), is only applicable in the case of the thermal (Gibbs) distribution of Bethe holes. A natural question here is whether there exist a general prescription which would permits to evaluate the thermodynamic Kohn's formula for generic equilibrium ensembles?

[6]

- Missing hyperlink in the reference to Appendix B.

- In the spirit of the above remarks, Eq.(6.2) which involves both the thermodynamic Y-functions and length L explicitly. The analysis there appears to crucially rely on the Cauchy's theorem, which requires $Y^{th}_j$ to have isolated poles in the physical domain. But since according to Eq.(5.1) $\zeta^k_j$ approach each other as $1/L$, I presume that poles of $Y^{th}_j$ are no longer isolated after taking the limit. How exactly does the logic work?

- "As shown in the previous subsection, the Drude weight D for arbitrary system size L is given by (6.6), and converges in the thermodynamic limit $L->\infty$ to the result derived by Zotos (6.18)."

How come the formula (6.6) is exact for any L given that it need the thermal distribution of holes as an input?

- "For finite size L we use discrete distributions that approximate the continuous densities as closely as possible."

Could the authors clarify how is this step implemented in practice? In my (possibly to naive) understanding, a simple sampling of thermal hole densities will typical violate the "on-shell" condition (4.3). I can imagine however that any explicit reference to quantization equations will render the whole analysis immediately infeasible. I hope that the authors can expand on this since this step appears to be quite vital to the main result of the paper.

[7]

- "the Drude weight exhibits a fractal dependence on $\Delta$"

I can recommend to include here also ref. [22], which first establishes saturation of the "fractal bound" of ref. [19] by invoking the complete quasi-particle spectrum.

- "For more quantitative and rigorous analysis of this intriguing behavior, a formula describing the Drude weight for any irrational numbers is highly desired."

I wonder why the authors chose to limit themselves to the restricted discrete set of simple roots of unity, given that the form of the Y-system for generic points is well known.

Requested changes

1- Please address the remarks and provide additional clarifications.

---

## Editorial Decision

editor-in-charge_assigned